# Effectiveness of an Intervention Programme on Adherence to the Mediterranean Diet in a Preschool Child: A Randomised Controlled Trial

**DOI:** 10.3390/nu14081536

**Published:** 2022-04-07

**Authors:** María Cristina Martíncrespo-Blanco, David Varillas-Delgado, Saray Blanco-Abril, María Gema Cid-Exposito, Juana Robledo-Martín

**Affiliations:** 1Nursing Research Unit, Hospital Universitario de Mostoles, 28935 Madrid, Spain; cmartincrespo@yahoo.es (M.C.M.-B.); saray.blanco@urjc.es (S.B.-A.); 2Department of Nursing, School of Health Sciences, Universidad Rey Juan Carlos, 28922 Madrid, Spain; gema.cid@urjc.es; 3Exercise and Sport Sciences, Faculty of Health Sciences, Universidad Francisco de Vitoria, 28223 Madrid, Spain; 4Gregorio Marañón Health Research Institute (IiSGM), Universidad Autonoma de Madrid, 28049 Madrid, Spain; juana.robledo@uam.es

**Keywords:** child, pre-school, primary prevention, child overnutrition, diet, Mediterranean, health child service

## Abstract

Background: The Mediterranean diet is considered one of the dietary patterns with the most accumulated scientific evidence on health benefits. In children, it has positive effects in the prevention of obesity and cardiovascular diseases, as well as in the prevention of diabetes. We aimed to evaluate the medium-term efficacy of an intervention programme, targeting adherence to the Mediterranean diet among preschool children. Methods: In a randomised, parallel trial of participants aged 3–5 years, a school garden was attended in the experimental group, and in the control group, the usual content on the human body and health were taught. Adherence to the Mediterranean diet was assessed using the KIDMED questionnaire, controlling for weight, height, body mass index (BMI) and socio-demographic variables. Results: A reduction in BMI was found in the experimental group after one year and at the end of the follow-up period. In the overall score obtained in the KIDMED survey, a statistical trend was found between the two groups (*p* = 0.076). In multivariate analysis, consumption of pulses more than once a week’ was predictive of improved diet quality, with an Odds Ratio (OR) in the experimental group of 1.382 (95% CI 1.126–1.695; *p* = 0.009). Conclusions: The experimental approach improved the quality of the participants’ diet, increasing adherence to the Mediterranean diet due to increased consumption of plant-based protein.

## 1. Introduction

The Mediterranean diet has been studied since the 1950s, when Keys compared the dietary habits of different cohorts from seven countries (USA, Japan, Finland, The Netherlands, the former Yugoslavia, Italy and Greece) and after a follow-up of 5–15 years, confirmed lower overall and coronary heart disease mortality and longer life expectancy in Mediterranean countries [1,2]. It is currently considered one of the dietary patterns with the most accumulated scientific evidence on health benefits [3], being declared Intangible Cultural Heritage of Humanity in 2010 [4].

Childhood obesity is a major public health priority [5] due to its increasing global presence at young ages [6] and its serious impact on morbidity in adulthood [7].

Its consequences are also felt during childhood, as children with obesity suffer from osteoarticular [8], cardiovascular [9,10], diabetes mellitus [11], tumours [12], behavioural [13] and psychosocial problems [14]. Although prenatal risk factors have been described in its aetiology [15], obesity is largely caused by living in an “obesogenic environment”, which includes socio-economic background, physical activity and eating habits [16].

Protective factors have also been identified, such as the healthy eating habits of parents [17,18] and consumption of a Mediterranean diet. This has positive effects on the prevention of obesity and cardiovascular disease among children, as well as the prevention of diabetes [19].

This diet is not unique, but is adapted to each geographical area [20] and is, thus, defined in terms of its nutritional composition, based on the consumption of mostly slow-acting carbohydrates, monounsaturated fatty acids, plant-based proteins, fibre and antioxidants [21].

Interventions developed to address childhood obesity through primary prevention can be classified into four broad groups: those focusing on diet-related issues; those promoting physical activity; those based on promoting behavioural change and multicomponent approaches [22]. Three validated questionnaires are used to analyse adherence to the Mediterranean Diet in children and adolescents: the Mediterranean Diet Adherence Questionnaire (KIDMED) [23], the Mediterranean Diet Score (MDS) [24] and the MDS-based food consumption frequency (fMDS) [25].

The KIDMED index [23], developed in the Spanish population for ages 2–24 years, is the most widely used to date. In a review published in 2017 that included 58 intervention studies, 27 of them used the KIDMED questionnaire directly as an instrument and another 16 studies used it indirectly through a food consumption register. Only 14 studies used the MDS and one study used the fMDS [21].

The effectiveness of interventions differs according to the type of approach and age group targeted. Among children aged 0–5 years, interventions aimed at promoting physical activity show moderate efficacy, but we do not yet have sufficient data to analyse the efficacy of those targeting diet alone [26], nor to analyse the absence of possible adverse effects of interventions, nor their long-term efficacy, as most studies published to date include a maximum follow-up period of 12 months [22,26]. 

The increase in childhood obesity and the need to promote proper nutrition from an early age means that new interventions are being considered. In this sense, recent reviews point out that interventions to promote healthy eating aimed at children should seek to awaken the interest of pupils, away from traditional education, with the school garden being a good resource for this purpose [27,28].

The implementation of school gardens provides a variety of benefits for children, such as increasing environmental awareness, encouraging teamwork, increasing motivation and promoting healthy eating habits, with the changes brought about in the eating habits and nutritional health of students being the object of study of most of the research carried out in several countries [29,30,31].

Among the benefits reported due to their use, Norman et al. [32] highlighted that they promote the students’ knowledge of unfamiliar foods, provide the educational centre with a natural space that allows them to learn knowledge and values related to nutrition, improve the students’ diet and at the same time reduce the problems derived from obesity.

The aim of the study was to evaluate the effectiveness of a school garden-based intervention in a pre-school child, aimed at adherence to the Mediterranean diet, including a long-term follow-up period of three years, and to analyse which factors determine greater adherence to this healthy eating pattern as body mass index (BMI) decreases.

## 2. Materials and Methods

### 2.1. Participants

The target population consisted of two groups of 3-year-old children enrolled in public pre-school and primary schools randomly selected in the city of Móstoles, Madrid, Spain.

In both groups, children suffering from chronic diseases or receiving corticosteroid treatment were excluded, as this could interfere with somatometry. Children whose parents did not understand spoken or written Spanish were also excluded.

Of the 138 children who were screened for eligibility from the two selected schools, two did not meet the inclusion criteria and were excluded. The remaining 136 participants were randomly assigned to the experimental and control groups (n = 68 in each group). Finally, 133 participants completed the study, excluding 3 who did not complete the entire follow-up process and who were missing weight and height data for some of the measurements: 65 in the experimental group and 68 in the control group (Figure 1).

Cluster randomisation was performed by dividing into two randomisation units, using a table of random numbers obtained from a free Excel software programme. The first randomisation unit comprised the educational districts into which the city was divided. Once one of the six zones was selected, a second randomisation process was applied to the list of all public schools in the zone. Two schools were selected: the first was the experimental group and the second the control group, and all children attending pre-school in the selected schools were included. Both schools are within 2 km of each other. The head teachers of these schools were contacted to see if they wanted to participate and to seek their permission. Each school received information about the objectives of this research and what their participation would entail, a schedule of activities and the variables to be measured, but they did not know what the experimental intervention consisted of, so they did not know whether they were in the control or experimental group.

Parents of the children signed informed consent prior to their participation in the study, protocol approved by the Clinical Research Ethics Committee of the Hospital Universitario de Móstoles (2016/013), Móstoles, Spain. The protocol and trial were conducted in accordance with the Declaration of Helsinki 1964 (last updated 2013) and the Good Clinical Practice Guidelines of the International Conference on Harmonisation (GCP ICH) and reported according to the CONSORT criteria. The trial was registered on ClinicalTrials.gov: NCT04301180.

### 2.2. Study Design

A randomized, parallel clinical trial was performed. Participants were randomly assigned to one of the two intervention groups: experimental and control groups of the corresponding beverage for 12 weeks.

The intervention programme ran concurrently with both groups over 10 weeks, in the last term of the school year, starting in April 2017 and with annual follow-ups also in the month of April over the next two years, to complete a monitoring period of 36 months. 

A randomised, parallel clinical trial was conducted. Participants were randomly assigned to one of two intervention groups: experimental group and control group for 12 weeks.

The intervention programme ran concurrently with both groups for 10 weeks, in the last term of the school year, starting in April 2017 and with annual follow-ups also in April for the following two years, to complete a 36-month follow-up period. 

In the first week, we collected information from both groups on the children’s age, weight and height, and subsequently calculated their body mass index (BMI). We also collected the questionnaires assessing the level of adherence to the Mediterranean diet (KIDMED), which assesses the consumption of milk and its derivatives, vegetables, fruits, legumes, fast food, sweets, soft drinks, meat and fish. Together with the above data, the socio-demographic variables of the parents or guardians were obtained by means of a data collection diary that had been provided the previous week to the families of all the children in both schools.

For the remaining nine weeks, in sessions of 40 min per week, the control group received the usual instruction corresponding to the human body and health module. In the experimental group, in 45-min sessions per week, all children in the school participated in a school garden care activity, which consisted of cleaning and preparing the soil and then planting seeds of different seasonal vegetables, observing their growth and taking care of the maintenance, watering and cleaning of the plot. During the work in the garden, we talked to the children about how food is produced, what types of food we eat and the healthy food pyramid. They explained where fruit and vegetables are found, their importance in our daily diet and how much we should eat, focusing also on the origin of milk, fish, meat and pulses and how often we should eat each of them. They worked on the importance of having a varied diet, how many meals a day and what they are called. During the work in the garden, they also talked about the composition of breakfast, hygiene when preparing food and hygiene when eating it, as well as fast food. After each of the sessions held with the children at school, they were given an information sheet for their parents on the topic they had dealt with that day. Along with this information, they were also given a worksheet with different drawing activities to do at home with the rest of their family. The information sheets and worksheets used the food pyramid as a basis: the first one contained information for parents on the pyramid and a drawing to colour together with the children; in the next one the information focused on the necessary water consumption accompanied by drawings to cut out with the children regarding hydration according to the weather conditions and thus start decorating the pyramid. In the following sessions information was given on each step of the pyramid accompanied by drawings to colour, cut out and paste with the children at each level until the pyramid was completed. Finally, the pyramids were laminated in order to have an individual breakfast tablecloth.

### 2.3. Main Outcome Measures

The main variable was the degree of adherence to the Mediterranean diet, and for this purpose the parents or guardians of the children in both groups were provided with a copy of the KIDMED questionnaire or Mediterranean Diet Adherence Test. This instrument assesses the degree of adherence to the Mediterranean diet in the population aged 2 to 20 years [33]. 

The sum of the values obtained in the KIDMED survey were classified into three levels: ≤3, diet of very low quality; 4–7, dietary patterns require improvement; ≥8, diet of very good quality in the Mediterranean diet [33]. The test questions were evaluated independently to determine predictors of good adherence to the Mediterranean diet.

### 2.4. Secondary Outcomes

Data were collected on children’s age (months), weight (kg) and height (cm), as well as socio-demographic information on parents: age (years), profession, ethnicity, education level and socio-economic status of both parents.

Data collection was carried out simultaneously in both groups. Parents received a data collection diary the week before the day set for measuring the children’s height and weight. Teachers were responsible for delivering and collecting these diaries after one week, together with the informed consent form allowing the child to participate in the study. To ensure that all documents were collected, parents were informed that they had to hand in the diary together with the informed consent form, indicating whether they authorised their child to participate in the study.

### 2.5. Sample Size and Power Analyses

A sample size of 130 children was calculated, assuming an expected dropout rate of 20% and a type I error of 0.005 (two-sided), which allows at least 80% power for the detection of statistically significant differences in the 4-point standard deviation on the KIDMED questionnaire [34].

### 2.6. Statistical Analysis

Quantitative variables were described by their mean values and standard deviation (SD), and qualitative variables by frequencies and percentages. The normality of the variables was assessed using the Kolmogorov–Smirnov test.

Analyses were performed per protocol. A homogeneity analysis was performed between the control and experimental groups, as well as in relation to the socio-demographic variables of the parents at the beginning of the study. Subsequently, the KIDMED score at baseline was compared with the score at each year. Statistical comparisons were made using the Student t-test or the Chi-square test if appropriate. Finally, all test questions were analysed independently by multivariate analysis, estimating risk by Odds Ratios (OR) and 95% confidence interval (95% CI) by logistic regression.

Statistical significance was defined as *p* ≤ 0.05 for a two-sided test. Analyses were performed with SPSS for Windows, version 21 (IBM Corp., Armonk, NY, USA). All data were analysed according to the pre-established protocol.

## 3. Results

### 3.1. Somatometric Characteristics

Children’s somatometric variables were measured at baseline, showing homogeneity between the study groups (Table 1), which were also homogeneous in terms of parents’ socio-demographic variables (Appendix A).

The evolution of children’s BMI was analysed over the course of the three-year intervention, both in the control group and in the experimental group, obtaining a statistically significant reduction in BMI in the experimental group after one year and at the end of the follow-up period (Table 2).

### 3.2. KIDMED Evaluation

The overall assessment of the KIDMED survey showed no differences at baseline between the study groups (*p* = 0.459), nor after one year of follow-up (*p* = 0.367), but we did find a statistical trend at the end of the follow-up period, with the experimental group obtaining a more acceptable mean score (6.48) than the control group (6.08) (*p* = 0.076). 

The KIDMED scale comprised sixteen questions, independently assessed to predict the overall effect of the test and to determine which items predicted good adherence to a healthy diet in both groups. Differences were found in three of them at the one-year follow-up, with a higher weekly consumption of pulses and nuts in the experimental group and a higher daily consumption of vegetables in the control group. At the end of the follow-up period, the difference in the consumption of legumes more than once a week was maintained in the experimental group, with a tendency to decrease the consumption of milk at breakfast in this group, while the consumption of vegetables was still higher in the control group, as well as a lower consumption of nuts and dried fruit, although the differences in these variables at 36 months were not significant (Table 3).

To analyse the predictor variables in the improvement of diet quality among the participating children in the two groups, a regression of the overall values of the KIDMED survey, as well as its individual questions—consume fresh or cooked vegetables once a day, consume pulses more than once a week, consume nuts at least 2–3 times a week, and drink milk and/or dairy products at breakfast—was performed, showing that the variable that predicts a statistical constant (*p* < 0.001) is the KIDMED survey question on consumption of pulses more than once a week at the end of the follow-up period, with an Odds Ratio (OR) for pulse consumption in the experimental group of 1.382 (95% CI 1.126–1.695; *p* = 0.009) compared to the control group.

## 4. Discussion

In this clinical trial, we obtained a prevalence of overweight in our sample population of 27.2%, the same as the figure obtained in the Child Nutrition Survey, carried out by the Madrid Regional Ministry for children aged 5 to 12 years, and the figure calculated by the World Health Organisation (WHO) at world level, which estimates a prevalence of overweight of 30%, exceeded in children under five years of age in developing countries [35]. 

However, in our study, we obtained a higher value than national values, where the prevalence of overweight in children aged 3–5 years in Spain is 21% [36] and that calculated by UNICEF/WHO and the World Bank for the Europe and Central Asia region, which estimates an overweight prevalence between 8.2 and 14.9% [37], which is consistent with data from the ENERGY project in the European Union [38]. This leads us to consider the influence of sociodemographic characteristics on diet [39].

The main contribution of this study is to identify the consumption of legumes more than once a week, as a predictor of good diet quality among preschool children (3–5 years), which is in line with the American Heart Association’s recommendations for increased consumption of plant-based proteins and substitution of red meat consumption to reduce cardiovascular risk [40]. Here there is also a trend towards lower consumption of nuts and higher consumption of fresh, cooked vegetables, milk and milk products in the control group. These results suggest that protein consumption in the experimental group changed, with an increase in the consumption of plant proteins and a decrease in the consumption of animal proteins from milk.

Protein intake recommendations for pre-school children indicate that protein should represent 10–15% of daily calorie intake. The most suitable sources are eggs and milk due to their high content of essential amino acids, followed by cereals and legumes as sources of plant-based protein [41]. At the start of the study, protein intake was higher than the amount recommended by the Spanish Society of Community Nutrition [42] and this was due to the intake of animal protein, a finding also indicated in the Child Nutrition Survey conducted by the Community of Madrid in children aged 5–12 years [36] and the nationwide EsNuPi study, which details how protein consumption was mainly made up of animal proteins, with pulses accounting for only 2.6% of daily energy intake [43]. The intervention carried out has been shown to be effective in increasing legume consumption more than twice a week, which would improve diet quality in this population group, since, as previous studies indicate, increasing vegetable protein intake positively influences diet quality in the pre-school population. Having identified animal protein consumption as a risk factor for weight gain, generally consumed through milk [44], the KIDMED only includes milk and fish among the protein sources assessed, so we cannot assess whether the consumption of meat and eggs has been modified after the intervention and whether the increased consumption of legumes has modified the consumption of animal proteins from these foods; it would be useful for future research to include, together with the test of adherence to the Mediterranean diet, a food consumption frequency questionnaire.

Comparing the results obtained with the food intake recorded by the 2001/2002 Madrid Child Nutrition Survey (ENICM) [36], the consumption of more than one serving of legumes increased in the experimental group and was maintained for 36 months (from 68.3% to 88.5%). This means that vegetable protein intake increased by one serving per week, so that children in the experimental group increased their intake from one serving per week at the beginning of the study to two servings per week at the end.

Regarding daily consumption of fresh or cooked vegetables, an indicator of dietary quality [45], it is interesting to note that in the experimental group, consumption decreased during the first year and that at the end of the follow-up, although a slight increase was observed, consumption was still lower than in the control group. However, in both groups, the values presented are higher than those indicated in the National Survey on Nutrition in the Child and Adolescent Population (ENALIA), which indicates a consumption of 25.4% of vegetables and salads in children aged 3–9 years [46].

The quality of the diet, considered to be close to the Mediterranean diet, shows a tendency to improve in the experimental group. However, its values, as in previous research, are not at optimal levels [33,47].

Maternal age has been identified as a conditioning factor in the adequacy of children’s diet at pre-school age, since at this age, food habits depend on the mother’s decisions [37]. In our study, there were no significant differences between the control group and the experimental group, so we can assume that this variable did not influence the results [48]. Different studies have also shown that having an unfavourable socioeconomic level and belonging to immigrant families are risk factors for childhood obesity and overweight [39,49,50]. Further, in our study, the control and experimental groups were homogeneous in these variables, so we can assume that they had no influence on the results obtained.

There is some controversy about the best place to deliver primary prevention interventions for childhood obesity in the preschool population, with some studies finding good results in the school [51], while other research shows greater efficacy of interventions delivered in the home, hospital or community [52]. Although most children of this age are in school, the most common route to reach them and their families is through the paediatrician’s office, in contrast to children aged 6 years and older, who are usually reached through their school.

Access to the study population involved coordination with school principals and teachers throughout the study. There was a loss of cases during the course of the research due to children moving to another school or being absent on the day the anthropometric measurements were taken. However, the percentage lost during the follow-up period was very low, and we do not believe that this caused a bias in the research, recognising schools as an appropriate means of accessing the target population.

This study evaluated a primary prevention intervention for childhood obesity in children aged 3–6 years, in a population with medium–low purchasing power. These are some of the strengths of its design, as most primary prevention interventions for childhood obesity do not target the 3- to 6-year-old population [52,53], along with the higher risk of overweight and obesity in children of immigrant parents and low socioeconomic status [54,55]. However, the generalisability of the results to groups with similar socio-demographic characteristics must be taken into account.

A particularly noteworthy feature of this research is the fact that the follow-up period lasted for 36 months, which made it possible to observe variations in their diet over time, a particularly important factor in this age group, as this is the time when dietary patterns are established [56].

Nut consumption, although it cannot be considered a determinant variable in diet quality, has shown a favourable trend as a predictor variable. This could be due to the sample size of the study and further research with a larger sample size is recommended to determine its influence as a predictor of better-quality nutrition in children.

It should be noted that the use of the KIDMED test on three occasions could lead to a recall bias; however, the fact that it was used with the same frequency in the control group and in the experimental group allows us to assume that the recall effect would have been similar in both groups.

The intervention programme was carried out simultaneously in two settings: the school and the home environment. Diet quality increases when interventions involve the whole family, as this is the most effective way for parents to make changes in their eating habits [53,57,58,59]. In our study, we did not control for family involvement, which prevents us from assessing the possible influence of this factor on the results.

The study presents several limitations: (1) It is necessary to consider, in both groups, the possible coexistence of this intervention with other simultaneous measures, since, due to the concern generated by this health problem, programmes are regularly implemented in schools to promote healthy breakfasts, for example, or involving the incorporation of nutritionists to prepare school lunch menus. (2) The short distance between the two schools located in the same residential area could have led to an exchange of information on both interventions between the participants. (3) The results of this questionable approach in such small groups are exaggerated. For this reason, research with larger sample sizes is needed to assess whether variables that show a trend, such as nut consumption and decreased milk consumption, may be determinants of diet quality.

## 5. Conclusions

The school garden intervention has been shown to be effective in reducing BMI in pre-school children aged 3–5 years, from low and middle socio-economic backgrounds.

There has been an increase in the consumption of plant-based proteins, a trend of decreasing consumption of animal protein from milk. Furthermore, this intervention increased the consumption of nuts.

The consumption of pulses more than once a week is a predictor of good diet quality at preschool age.

## Figures and Tables

**Figure 1 nutrients-14-01536-f001:**
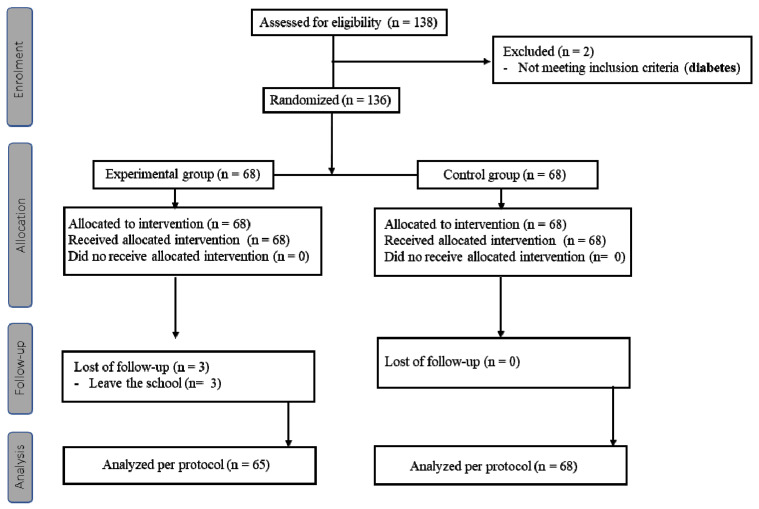
Flow chart.

**Table 1 nutrients-14-01536-t001:** Homogeneity analysis between the study groups.

		Experimental Group (n = 65)	Control Group (n = 68)	*p* Value
Age, years (SD)		48.47 (2.837)	47.86 (2.888)	0.24
Sex, n (%)	Boys	33 (55.0)	34 (50.7)	0.632
	Girls	27 (45.0)	33 (49.3)	
Weight, Kg (SD)		18.023 (2.226)	18.075 (1.874)	0.887
BMI, n (%)	Normal	35 (58.3)	45 (69.2)	0.370
	Overweight	18 (30.0)	16 (24.6)	
	Obesity	7 (11.7)	4 (6.2)	
Height, cm (SD)		104.16 (5.359)	105.16 (4.106)	0.246
BMI, Kg/m^2^ (SD)		16.55 (1.199)	16.31 (1.131)	0.247

BMI, Body mass index; cm, centimetres; Kg, kilograms; m, meters; SD, Standard deviation.

**Table 2 nutrients-14-01536-t002:** Evolution of BMI values in the study groups.

	Groups	*n*	Mean (SD)	*p* Value
BMI at the start of the study	Experimental	60	16.5570 (1.19939)	0.247
	Control	65	16.3143 (1.13131)	
BMI after 12 months of monitoring	Experimental	59	16.12004 (1.12004)	0.026
	Control	63	16.07826 (1.07926)	
BMI after 36 months	Experimental	61	15.7430 (1.29284)	<0.001
	Control	64	16.5111 (1.06097)	

BMI, Body mass index; SD, Standard deviation.

**Table 3 nutrients-14-01536-t003:** KIDMED predictor variables.

	Beginning	12 Months	End of Follow-Up
Total KIDMED score	Experimental Group: 5.90	Experimental Group: 6.09	Experimental Group: 6.48
	Control Group: 6.08	Control Group: 5.88	Control Group: 6.08
	*p* = 0.459	*p* = 0.367	*p* = 0.076
Eats fresh or cooked vegetables at least once a day	Experimental Group: 43.3%	Experimental Group: 40.7%,	Experimental Group: 44.3%
	Control Group: 50.8%	Control Group: 64.1%	Control Group: 60.9%
	*p* = 0.405	*p* = 0.009	*p* = 0.063
Consumes legumes more than once a week	Experimental Group: 68.3%	Experimental Group:88.9%	Experimental Group:88.5%
	Control Group: 67.7%	Control Group:64.1%	Control Group:64.1%
	*p* = 0.939	*p* = 0.002	*p* = 0.001
Consumes dried fruit and nuts at least 2–3 times a week	Experimental Group: 73.3%	Experimental Group: 86.4%	Experimental Group: 83.6%
	Control Group: 70.8%	Control Group: 70.3%	Control Group: 70.3%
	*p* = 0.750	*p* = 0.031	*p* = 0.078
Has milk and/or dairy for breakfast	Experimental Group: 86.7%	Experimental Group: 86.4%	Experimental Group: 72.1%
	Control Group: 86.2%	Control Group: 85.9%	Control Group: 84.4%
	*p* = 0.930	*p* = 0.933	*p* = 0.096

## Data Availability

Not applicable.

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
