# Peer review of "Effectiveness of an Intervention Programme on Adherence to the Mediterranean Diet in a Preschool Child: A Randomised Controlled Trial"

_nutrients, 2022, doi:10.3390/nu14081536_

Round 1

Reviewer 1 Report

Very interesting study, thank you, I recommended a couple of areas for more clarity as can be seen in the attached manuscript. Particularly in the Methods section.  English should be reviewed by a langauge editor. 

Author Response

Response to Reviewers’ comments

nutrients-1609803

Effectiveness of an intervention program on adherence to the Mediterranean diet in a pre-school child: a randomized controlled trial

We sincerely thank the Reviewer for carefully proofreading the manuscript and for their helpful and constructive comments. We have addressed all the points raised by the Reviewers in this response letter and we have highlighted any change to manuscript in red using the track changes tool. We believe that our manuscript has been improved by the suggested changes.

Reviewer: 1

Very interesting study, thank you, I recommended a couple of areas for more clarity as can be seen in the attached manuscript. Particularly in the Methods section. English should be reviewed by a language editor.

Thanks for the expert considerations of the reviewer in the manuscript. The authors have improved the writing of material and methods for a better understanding according to the suggestions shown.

There is something missing here, we collected data from 3-year-old children... or Our population groups comprised of 3-year-old children...

Thanks for this suggestion. Following the reviewer recommendations, it has been specified that the target population was made up of 3-year-old children from the selected infant education schools.

How far apart were these schools, did the children and or parents from the 2 schools not come in to contact with each other which could have impacted on their knowledge and or influence their answering

The distance to which the selected public schools are located (2 km) has been specified, indicating in the limitations of the study that we cannot ensure that the parents of the two schools did not have contact with each other outside the school and therefore knowledge of the intervention that was being carried out in the other school which may have influenced their knowledge and their answers.

Would be good to have a little more information about the roll out and detail of the gardening, was it vegetable gardens or other and what did they do?

Following the recommendations of the reviewer, we have detailed the activities in which the children participated in the school garden (cleaning and preparing the soil and then planting seeds of different seasonal vegetables, observing their growth and taking care of the maintenance, watering and cleaning of the soil) as well as the other activities that took place in the intervention group.

A little more detail about the healthy eating guidance took place or advice given to the children and parents, what was included in the information sheet for the parents.

Thank you for this suggestion. The authors have modified the wording for better understanding as follows; "The information sheets and worksheets used the food pyramid as a basis: the first one contained information for parents on the pyramid and a drawing to colour together with the children, in the next one the information focused on the necessary water consumption accompanied by drawings to cut out with the children regarding hydration according to the weather conditions and thus start decorating the pyramid, in the following sessions information was given on each step of the pyramid accompanied by drawings to colour, cut out and paste with the children at each level until the pyramid was completed. Finally, the pyramids were laminated in order to have an individual breakfast tablecloth".

Please explain how this worked, you indicated earlier on that from the 2 schools one school was control and one the experimental group, so was it 138 per school, or 138 across the 2 schools?

Thanks for this suggestion. According to reviewer consideration, has been specified that initially a total of 138 children belonging to the two selected schools were selected.

It is not clear in the start of the paper that consumption of legumes more than once a week will be used as predictor variable, it should bes described in the methods section

Thanks to the reviewer for this suggestion. It has been specified in the Materials & Methods section for better understanding.

On behalf of all co-authors, many thanks for the insightful review.

Reviewer 2 Report

GENERAL COMMENT

Thank you for giving me the possibility to comment on your paper.

Analysing whether nutrition education interventions in nursery schools as a measure to prevent the development of obesity and achieve an improvement in eating habits is interesting and relevant, however there are several sections of the article that need to be improved.

SPECIFIC COMMENTS

Introduction:

Introduction should be improved focusing on the topics studied.

Results

Section 3.1 should be relocated to the material and methods section.

It should be kept in mind throughout the article that the aim of the study was to evaluate the effectiveness of an intervention program in 64 a pre-school child, targeting their adherence to the Mediterranean diet, including a 65 long-term monitoring period of three years, and to analyse which factors determine 66 greater adherence to this healthy eating pattern, as the BMI decrease

Table 2 should be included as supplementary material.

Sections 3.3 and 3.4 should be unified as they analyse the results obtained with the KIDMED test.

Discussion:

The conclusions on the dietary variations obtained are not supported by the stated results of the study. This should be modified.

The discussion should be improved by analysing the results obtained with respect to similar results in other countries. It is important add the limitations of your work and analyse why virtually no changes were obtained in the diet of the intervention group.

Conclusions:

There is a need to improve the conclusions by focusing on the results obtained.

Author Response

Response to Reviewers’ comments

nutrients-1609803

Effectiveness of an intervention program on adherence to the Mediterranean diet in a pre-school child: a randomized controlled trial

We sincerely thank the Reviewer for carefully proofreading the manuscript and for their helpful and constructive comments. We have addressed all the points raised by the Reviewers in this response letter and we have highlighted any change to manuscript in red using the track changes tool. We believe that our manuscript has been improved by the suggested changes.

Reviewer 2

GENERAL COMMENT

Thank you for giving me the possibility to comment on your paper.

Analyzing whether nutrition education interventions in nursery schools as a measure to prevent the development of obesity and achieve an improvement in eating habits is interesting and relevant, however there are several sections of the article that need to be improved.

SPECIFIC COMMENTS

Introduction:

Introduction should be improved focusing on the topics studied.

Thanks for this suggestion. According to the expert consideration of the reviewer, the introduction has been improved by focusing on the topics studied and expanding the information on the relevant points of the study.

Results

Section 3.1 should be relocated to the material and methods section.

Thanks for this suggestion. According to reviewer consideration, this section has been relocated to material and methods section, renamed 2.1 Participants

Table 2 should be included as supplementary material.

According to reviewer suggestion, Table 2 has been included as supplementary material an renamed as Suppl material. Tables of the manuscript has been renamed.

Sections 3.3 and 3.4 should be unified as they analyse the results obtained with the KIDMED test.

Thanks for this suggestion. Sections 3.3 and 3.4 has been unified in section 3.3 KIDMED evaluation for better understanding according reviewer consideration.

Discussion:

The conclusions on the dietary variations obtained are not supported by the stated results of the study. This should be modified.

According to reviewer suggestion, the conclusion has been modified to support the results of the study.

The discussion should be improved by analysing the results obtained with respect to similar results in other countries. It is important add the limitations of your work and analyse why virtually no changes were obtained in the diet of the intervention group.

Thanks for this suggestion. According to the review improvement, has been added analysis of the results obtained comparing similar results. Limitations of the study has been added and explained why virtually no changes were obtained in the diet of the intervention group.

Conclusions:

There is a need to improve the conclusions by focusing on the results obtained.

Thanks for this suggestion. According to reviewer suggestion, the conclusion has been modified to support the results of the study.

On behalf of all co-authors, many thanks for the insightful review.

Round 2

Reviewer 2 Report

It is appreciated that the authors have implemented the requested improvements, however they should highlight in the results and discuss in addition to the improvements in the consumption of legumes the low consumption of fresh or cooked vegetables in the experimental group, as this is a clear indicator of dietary quality that has not been increased despite the intervention carried out. On the other hand, as the consumption of other animal proteins (meat, fish and eggs) has not been recorded/indicated, it cannot be affirmed that pulses replace their consumption, as on the contrary, it could be that they have increased their consumption at the expense of the low quantities of vegetables consumed.

This should be reflected in the article.

Author Response

Response to Reviewers’ comments

nutrients-1609803

Effectiveness of an intervention program on adherence to the Mediterranean diet in a pre-school child: a randomized controlled trial

We sincerely thank the Reviewer for carefully proofreading the manuscript and for their helpful and constructive comments. We have addressed all the points raised by the Reviewer in this response letter and we have highlighted any change to manuscript in red using the track changes tool. We believe that our manuscript has been improved by the suggested changes.

Reviewer 2

It is appreciated that the authors have implemented the requested improvements, however they should highlight in the results and discuss in addition to the improvements in the consumption of legumes the low consumption of fresh or cooked vegetables in the experimental group, as this is a clear indicator of dietary quality that has not been increased despite the intervention carried out. On the other hand, as the consumption of other animal proteins (meat, fish and eggs) has not been recorded/indicated, it cannot be affirmed that pulses replace their consumption, as on the contrary, it could be that they have increased their consumption at the expense of the low quantities of vegetables consumed.

This should be reflected in the article.

The authors are grateful for the reviewer's suggestion.

Based on the indications, the writing has been improved for a better understanding of what has been studied, showing the limitations of KIDMED to assess other animal proteins, suggesting future studies to use other questionnaires and increase the quality of the analysis of animal proteins.

On behalf of all co-authors, many thanks for the insightful review.
